# Amelioration of Mitochondrial Quality Control and Proteostasis by Natural Compounds in Parkinson’s Disease Models

**DOI:** 10.3390/ijms20205208

**Published:** 2019-10-21

**Authors:** Bongki Cho, Taeyun Kim, Yu-Jin Huh, Jaemin Lee, Yun-Il Lee

**Affiliations:** 1Division of Biotechnology, Daegu Gyeongbuk Institute of Science and Technology, Daegu 42988, Korea; cbk34@dgist.ac.kr; 2Department of New Biology, Daegu Gyeongbuk Institute of Science and Technology, Daegu 42988, Korea; xoxsdfaa7876@dgist.ac.kr (T.K.); yujin.huh@dgist.ac.kr (Y.-J.H.); 3Well Aging Research Center, Daegu Gyeongbuk Institute of Science and Technology, Daegu 42988, Korea

**Keywords:** Parkinson’s disease (PD), mitochondrial dysfunction, dynamics, hormesis, proteostasis, ubiquitin‒proteasome system (UPS), autophagy, mitophagy, natural compounds

## Abstract

Parkinson’s disease (PD) is a well-known age-related neurodegenerative disorder associated with longer lifespans and rapidly aging populations. The pathophysiological mechanism is a complex progress involving cellular damage such as mitochondrial dysfunction and protein homeostasis. Age-mediated degenerative neurological disorders can reduce the quality of life and also impose economic burdens. Currently, the common treatment is replacement with levodopa to address low dopamine levels; however, this does not halt the progression of PD and is associated with adverse effects, including dyskinesis. In addition, elderly patients can react negatively to treatment with synthetic neuroprotection agents. Recently, natural compounds such as phytochemicals with fewer side effects have been reported as candidate treatments of age-related neurodegenerative diseases. This review focuses on mitochondrial dysfunction, oxidative stress, hormesis, proteostasis, the ubiquitin‒proteasome system, and autophagy (mitophagy) to explain the neuroprotective effects of using natural products as a therapeutic strategy. We also summarize the efforts to use natural extracts to develop novel pharmacological candidates for treatment of age-related PD.

## 1. Introduction

Parkinson’s disease (PD) is the second most common neurodegenerative disease. Approximately 1% of the elderly population above 60 years of age suffers from PD, and the prevalence of the disease increases to 4% in the highest age group [1]. Because the incidence of PD depends strongly on age, the number of PD patients is estimated to dramatically increase as lifespans also increase. The economic burden of PD was estimated to be $14.4 billion in the United States in 2010 [2]. However, it increased up to $51.9 billion in 2017 [3], and is expected to increase more dramatically in the future. The most effective therapeutic option is the administration of l-3,4-dihydroxyphenylalanine (L-DOPA), which can cross the blood‒brain barrier and be metabolized to dopamine [4]. However, all currently available drugs, including L-DOPA, only modulate dopamine levels in PD patients’ brains and are of limited effectiveness in the initial stages of the disease, which can last for 1–5 years [5]. Novel strategies are therefore needed to prevent and manage PD in the later stages.

PD is histologically characterized by the progressive loss of dopaminergic (DA) neurons in the substantia nigra pars compacta (SNpc), which innervates basal ganglia and regulates motor control through the release of dopamine. The loss of DA neurons occurs before the onset of motor symptoms [6]. At the end stage of PD, neuronal degeneration become widespread, resulting in various symptoms. Another notable characteristic of PD is Lewy pathology (LP), particularly within the brain stem and olfactory system during early-stage PD. As the disease progresses, LP spreads to the limbic and neocortical regions of the brain. LP is usually observed in PD patients’ brains using histopathological methods [7]. However, LP is also observed in non-PD human brains, making LP a poor predictor of PD [8].

### 1.1. Major Pathological Mechanisms of Neurodegeneration in PD

The mechanism of PD pathogenesis has been studied extensively, although questions remain [9,10]. In brief, impairment of quality control in mitochondria and proteins by oxidative stress, and α-synuclein accumulation, is the primary mechanism associated with degeneration of DA neurons in PD with neuroinflammation [10]. Because this pathological mode is a common characteristic in other neurodegenerative diseases, including Alzheimer’s disease and amyotrophic lateral sclerosis, we will discuss PD-specific pathological mechanisms of mitochondrial quality control and proteostasis.

### 1.2. Impairment of Mitochondrial Quality Control

Several genes have been identified to be related with early-onset PD, and their physiological roles have been extensively studied. Parkin and PINK1 are major components for autophagy-mediated degradation of mitochondria (mitophagy), and their genetic mutations are closely related with accumulation of dysfunctional mitochondria in early-onset PD [11,12]. In addition, DJ-1 is critical for the antioxidant process against oxidative stress, which is induced by Ca^2+^ oscillation in autonomously pacemaking DA neurons [13,14], and its autosomal recessive mutation is also related with early-onset PD [15]. These observations suggest a pathological role of mitochondrial dysfunction in early-onset and potentially sporadic PD. Especially, decreased activity of mitochondrial respiratory chain complex I has been observed in post-mortem SNpc of sporadic PD patients [16]. Neurotoxins, such as 6-hydroxydopamine (6-OHDA), 1-methyl-4-phenyl-1,2,3,6-tetrahydropyridine/1-methyl-4-phenyl-pyridinium (MPTP/MPP^+^) and rotenone have been frequently used for experimental PD model. They inhibit the activity of mitochondrial respiratory chain complex I, and aberrantly induce mitochondrial dysfunction by oxidative stress, thereby mimicking selective loss of DA neurons in SNpc [17,18]. These indicate that impairment of mitochondrial function is linked with PD pathology.

Impairment on mitochondrial turnover also appears in PD [19]. Mitochondrial turnover is mediated by two pathways; 1) morphological balance between fusion and fission, and 2) qualitative and quantitative balance between biogenesis and mitophagy. Mitochondrial fragmentation has been well known as a common phenomenon in early stage of neuropathology including PD [20]. And reversely, mdivi-1, a synthetic blockade of mitochondrial fission as an inhibitor of Dynamin-related protein 1 (DRP1) [21], efficiently rescues DA neurons in a genetically- and chemically-induced PD model [22,23], emphasizing a critical contribution of mitochondrial dynamics in PD pathology. In addition, level of genes controlled by proliferator-activated receptor gamma coactivator 1-alpha (PGC1α), which is a master transcription factor for mitochondrial biogenesis, are downregulated in the brains of PD patients [24]. Reversely, activation of PGC-1α signaling efficiently reduces α-synuclein toxicity [25]. Furthermore, overexpression of Parkin prevented degeneration of DA neuron in PD model through activating mitophagy [26]. Those studies suggest that the activation of mitochondrial quality control can be a strategy to prevent and manage sporadic PD.

### 1.3. Impairment of Proteostasis

The second pathological mechanism of PD is abnormal accumulation of misfolded proteins by impairment of proteostasis. α-synuclein has been reported to be a major component in Lewy bodies in PD patients, and its mutation is involved in early-onset PD [10], raising the possibility that α-synuclein aggregates may play a critical role in PD pathogenesis. Although the physiological role of α-synuclein remains to be understood, the detrimental outcome of α-synuclein oligomers and aggregates has been widely studied. In pathological conditions, α-synuclein can oligomerize and form insoluble fibrils [27]. The α-synuclein oligomer induces aberrant generation of reactive oxygen species by inhibiting mitochondrial respiratory complex I, and leads to mitochondrial dysfunction [28]. Enhancement of proteostasis of α-synuclein by preventing aggregation and/or clearing aggregates can therefore be an effective strategy to cope with PD. A study in transgenic mice expressing human α-synuclein demonstrated that both the ubiquitin‒proteasome system (UPS) and autophagy‒lysosome pathway are responsible for the degradation of α-synuclein in neurons [29]. Rapamycin, an inhibitor of the mammalian target of rapamycin, consistently promotes degradation of wild-type and mutant α-synuclein [30] and rescues loss of DA neurons and parkinsonism in a 6-OHDA-induced PD mouse model [31]. These observations suggest that the activation of proteostasis mechanisms can be an effective strategy to manage PD through α-synuclein clearance.

Parkin plays a critical role in mitophagy [26] and gene transcription [32] as a PD-related multifunctional E3 ligase. Parkin targets, ubiquitinates, and degrades other proteins as well as the substrates involved in mitophagy. For instance, the genetic inactivation of Parkin leads to the accumulation of ZNF746 (PARIS), a substrate of Parkin, and this process represses PGC-1α signaling, leading to the degeneration of DA neurons [33]. PARIS accumulates excessively and consistently in familiar and sporadic PD patients’ brain, indicating a pathophysiological role in PD. Parkin also ubiquitinates and degrades the aminoacyl-tRNA synthetase complex interacting multifunctional protein-2, which activates poly(ADP-ribose) polymerase-1 and promotes PAR polymerization, resulting in neuronal death via “parthanatos” [34,35]. These studies suggest a crucial role for E3 ligase activity of Parkin in the PD-related degeneration of DA neurons. Activation of UPS by Parkin or other E3 ligase may therefore also offer a crucial neuroprotective effect against PD.

## 2. Compounds from Natural Products Alleviating Mitochondrial Dysfunction in PD

### 2.1. Recovery of Redox Homeostasis

We list 84 lead compounds isolated from natural products that have neuroprotective effect in vitro and/or in vivo experimental PD models according to their chemical class with effect summary (Table 1). Among them, the reaction of some natural compounds in mitochondrial quality control is summarized in Figure 1. Oxidative stress has been proposed as a main initial factor in mitochondrial dysfunction, which appears as an early pathological event in neurodegenerative diseases, including PD [36]. Mitochondria are the main endogenous source of various free radicals, including reactive oxygen species/reactive nitrogen species (ROS/RNS) via oxidative phosphorylation and are removed by redox enzymes including catalase, superoxide dismutase, and heme oxigenase-1 with intracellular antioxidants such as glutathione (GSH) [37]. However, the failure of redox homeostasis induces excessive levels of ROS/RNS, leading to mitochondrial dysfunction [36]. Neurotoxins in experimental PD models, such as 6-OHDA, MPP^+^/MPTP, rotenone, and paraquat, impair redox homeostasis by reducing the amount of antioxidants and activity of redox enzymes [38]. Traditionally, many compounds from natural products that recover redox homeostasis have been suggested for mitochondrial quality control in PD. Pre- or cotreatment of the compounds efficiently reduces levels of ROS/RNS against PD-related neurotoxins. Although the compounds, which are classified as polyphenols, terpenes, saponins, alkaloids, and other classes, exhibit anti-oxidizing activity in vitro, they may work as cellular activators and/or messengers by increasing the amount of GSH and by enhancing the activity of redox enzymes. Some mechanistic studies have revealed that nuclear factor erythroid 2-related factor 2 (NRF2) plays a central role in activating the redox system for neuroprotection against PD. Upon oxidative stress, NRF2 is stabilized by escaping from the UPS, which is mediated by Kelch-like ECH-associated protein 1 (KEAP1) and Cullin-3 (CUL3) [39]. Therefore, it accumulates in the nucleus and binds to promoters of multiple redox enzyme genes as a transcriptional activator, leading to the expression of redox enzymes as a defensive response. This process is enhanced by the following compounds: baicalein [40], luteolin [41], naringenin [42], puerarin [43] and genistein [44], auraptene [45], resveratrol [46], 11-dehydrosinulariolide [47], tanshinone I/IIA [48,49], astaxanthin [50], notoginsenoside Rg2/Rd/Re [51,52], ligustrazine [53], fucoidan [54], gastrodin [55], 3,4-dihydroxyphenyl-lactic acid [56], and salidroside [57]. However, some compounds induce expression of DJ-1, which promotes the recovery of the redox system via SOD1 and NRF2 signaling [58]. Among them are naringenin [59], sesamol [59], 11-dehydrosinulariolide [47], salidroside [57], rutin [60], and isoquercitrin [60]. Previous studies have demonstrated that various polyphenols and terpenes can evoke NRF2 signaling in other cellular contexts and environments [54]. This implies that other listed compounds can also activate NRF2 signaling, and their mechanistic study in PD models should be pursued. Taken together, we suggest that recovery of redox homeostasis is a basic property of natural compounds in PD treatment.

### 2.2. Enhancement of Mitochondrial Turnover by Structural Dynamics

Recent papers have revealed the importance of structural quality control of mitochondria in neurodegeneration, including PD [20]. In the intra-/extracellular environment, mitochondria undergo dynamic morphological changes via controlled fusion and fission, which are mediated by fusion proteins, mitofusin1/2 and optic atrophy 1 (OPA1), and the fission protein DRP1 [146]. This process contributes to mitochondrial quality and bioenergetics by the sharing and division of metabolites and nucleoids in mitochondria (Figure 1). However, PD-related neurotoxins and genetic mutations can induce excessive fragmentation of mitochondria by enhancing fission or inhibiting fusion, resulting in excessive mitophagy and eventual mitochondria-mediated neuronal death [19]. As a result of this discovery, compounds that inhibit mitochondrial fragmentation in PD models have been proposed. Thymoquinone reverts rotenone-induced upregulation of DRP1 protein in substantia nigra and striatum in PD model rats [126]. Rutin and isoquercitrin recover the expression of OPA1 in 6-OHDA-treated PC12 cells [60]. Moreover, other compounds promote mitochondrial turnover by enhancing the overall activity of fusion/fission or mitophagy. Resveratrol upregulates the expression of both MFN1/2 and DRP1, resulting in the upscaling of mitochondrial quality by enhanced fusion/fission of mitochondria in PD models [111,112]. Kaempferol induces mitochondrial fragmentation, which contributes to efficient mitophagy, thereby protecting neurons from accumulation of abnormal mitochondria [80]. Rosmarinic acid protects membrane integrity in mitochondria against permeabilization by α-synuclein aggregates [109].

### 2.3. Natural Compounds Evoking Mitochondrial Hormesis

Hormesis-evoking therapeutic trials in PD have been conducted because the pathology of sporadic PD is closely linked with mitochondrial aging [147]. Hormesis is an adaptive response against severe challenges by enhancing functionality and tolerance upon preconditioned mild intracellular or extra-environmental stress [148]. Especially, mitochondrial hormesis can be evoked in response to mild mitochondrial stressors, including energetic depletion, calcium, and ROS by adaptive endoplasmic reticulum (ER)/integrated stress response and mitochondrial unfolded protein response [149]. This process promotes biogenesis, energetics, antioxidant response, protein quality control, and mitophagy of mitochondria, thereby extending lifespans with reduced metabolism via cytokine-mediated systemic regulation. Treatment with epigallocatechin gallate [61], quercetin [73], resveratrol [113] or fucoidan [123], sesamol/sesamin [128], astragaloside IV [78], panaxatriol saponin [84], or salidroside [144] in PD models activates sirtuin 1 (SIRT1) signaling, which promotes PGC1α signaling and Forkhead box O3 signaling, which are involved in the biogenesis/bioenergetics and mitophagy/redox of mitochondria, respectively [149]. In addition, rutin and oleuropein upregulate IRE1α and ATF-4 without activating CHOP, PERK, BIP, and PDI in low hormetic doses, thereby improving cell survival [76]. However, relatively high doses of panaxatriol saponin, rutin, and oleuropein inhibit cell growth and proliferation, indicating some toxic effect. Therefore, these hormesis-evoking compounds may require more intensive study on the dose‒response [76,84]. SIRT1 signaling also activates the NRF2-mediated activation of the redox system via PGC1α signaling [149]. Therefore, NRF2-activating compounds may have a potential hormetic effect, but this possibility requires further study.

## 3. Natural Compounds Ameliorating Proteostasis Impairment in PD

The best-described pathological feature of PD is compromised proteostasis, which can be induced by oxidative or nitrosative stress resulting from misfolded protein accumulation and other exogenous neurotoxins [150,151]. In this section, we focus on two major mechanisms involved in proteostasis impairment with PD onset: UPS and autophagy. Autosomal recessive mutations of Parkin represent a large proportion of familial PD [152,153], and disruption of Parkin-mediated proteolysis leads to excessive protein misfolding, which culminates in PD [154]. On the other hand, α-synuclein forms fibril aggregates via PD-associated progressive posttranslational modifications, and it is usually degraded by autophagy‒lysosome machinery. However, pathologically excessive α-synuclein aggregates impair the autophagy‒lysosome machinery, leading to the vicious establishment of PD [155]. Researchers have therefore focused on ameliorating the collapsed protein quality for PD by controlling translation, chaperone-assisted folding and the degradation of protein. The regulation on proteostasis machinery by natural compounds is summarized in Figure 2.

### 3.1. Regulation through the Ubiquitin‒Proteasome System

One of the protein degradation pathways is UPS. Proteins are polyubiquitinated by E3 ligase and finally cleared by the proteasome. Some studies have tried to restore the impaired activity of UPS in PD models by using natural compounds. Salidroside decreases the level of phosphorylated α-synuclein (pSer129) by recovering proteasome activity in UPS-impaired PD models by 6-OHDA [145]. Because the E3 ligase, which catalyzes the polyubiquitination reaction, provides a key regulatory function in UPS, the regulation of its activity has been studied as a therapeutic strategy for PD. Some studies reported on the UPS-mediated regulation of p53, which is a key mediator of neuronal death in neurodegenerative diseases [156]. In PD patient brains, p53 is accumulated, and is involved in the degeneration of DA neurons [157]. Generally, MDM2, an E3 ligase, degrades p53, and could be activated by p53 in a negative feedback loop [158,159]. Upon cellular stress, including DNA damage, p53 becomes stable through its phosphorylation, mainly at the Ser-15 and -37 residues [160,161]. Due to its modification, the phosphorylated p53 destabilizes MDM2 and finally disorganizes the UPS function, leading to the aberrant protein accumulation. Some polyphenols, including flavonoids and lignans, have been reported to exhibit protective effect on impaired UPS regulating p53. Epigallocatechin gallate, rottlerin [62], puerarin [71], sesamol, and naringenin [59] inhibit the aberrant accumulation of p53 by recovering MDM2-mediated UPS, thereby suppressing p53-dependent cell death in PD models [62,71]. On the other hand, Parkin has an E3 ligase function, and its regulation has been investigated [162]. However, regulating Parkin activity through natural products is still under investigation. Another E3 ligase, IDUNA (RNF146), has PAR-dependent E3 ligase activity [163]. It protects against programmed cell death (called parthanatos) through proteasomal degradation. Recent studies have discovered that the natural products liquiritigenin and rhododendrin provide a neuroprotective effect in 6-OHDA PD models by inducing IDUNA activity. Both products bind to estrogen receptor-β stimulating transcription of IDUNA [97,164].

### 3.2. Regulation through the Autophagy‒Lysosomal Pathway

Another major protein degradation pathway is autophagy. It is a kind of pro-survival pathway, which clears misfolded or damaged proteins that cannot be degraded by unfolded protein response. Several toxin-induced PD models have been used to simulate the epidemiology of PD. Through exogenous toxins, ER stress evoked from increased ROS generation and decreased ATP synthesis can directly impair mitochondrial respiratory complex I [165]. Many studies have reported natural products that can treat these impaired mitochondrial environments by increasing autophagy flux and targeting specific mechanisms. A well-known natural product, quercetin, is an autophagy enhancer that plays a protective role in response to ER stress in rotenone-induced PD rat models. Quercetin treatment ameliorates DNA fragmentation and decreases beclin-1 levels [74]. Triptolide [68], Amurensin G [102] and celastrol [69] induces autophagy by activating LC3-II upregulation and clears α-synuclein in vitro and in vivo PD models. Some studies have reported that natural products can elevate autophagic activity through the modulation of AKT/AMPK/mTOR signaling. An oxindole alkaloid, corynoxine, has been described as an autophagy inducer. Chen et al. (2014) suggested that corynoxine-induced autophagy can clear α-synuclein through the Akt/mTOR pathway in neuronal cells and a *Drosophila* model [139]. Furthermore, Chen et al. (2017) introduced a model of corynoxine-induced neuronal autophagy. They established a network-based algorithm of in silico kinome activity profiling, and predict phosphoproteomic data. They then suggested that corynoxine-induced autophagy could clear α-synuclein regulated by MAP2K2 and PLK1 kinase activity [166]. Onjisaponin B derived from Radix Polygalae was reported to have regulatory function of autophagy, enhancing autophagy flux by the AMPK/mTOR signaling pathway and finally removing α-synuclein A53T mutant proteins [86].

### 3.3. Inhibition of Protein Aggregation Formation

The most frequently described protein in the pathology of PD is α-synuclein. Aggregates of α-synuclein can be toxic in cellular environments and can lead to PD [167]. Once α-synuclein forms a fibril structure, it cannot be easily degraded through the protein degradation pathway. Inhibition of the formation of α-synuclein aggregates is therefore a promising therapeutic strategy. Some studies have reported novel natural products that control α-synuclein oligomerization. In particular, the polyphenol family has demonstrated an ability to directly or indirectly inhibit α-synuclein oligomerization. Curcumin is a well-known antioxidant that can increase the solubility of the α-synuclein form of monomers in catecholaminergic cell lines and in vivo models, thereby inhibiting oligomerization [168,169,170,171]. Pretreatment of rosmarinic acid inhibits reduction in the mitochondrial membrane potential and α-synuclein aggregation through its iron-chelating activity in an MPTP-induced PD model [110]. In addition, myricetin can inhibit α-synuclein oligomerization by directly binding to the α-synuclein N-terminal region in vitro [90]. Tanshinone I and tanshinone IIA decreased the formation of α-synuclein oligomers [65]. Ginsenoside Rb1 dissociates α-synuclein fibrillation through directly binding to α-synuclein oligomers [88]. Tea polyphenols have been shown to protect DA neurons against PD in mice models. Additionally, their therapeutic effects have been reproduced in an MPTP-induced monkey PD model that prevents α-synuclein oligomerization [172].

## 4. Conclusions and Future Prospects

In this review, we discussed the neuroprotective effects of lead compounds from natural products on mitochondrial quality control and proteostasis in experimental PD models. Unlike synthetic drugs that target only single molecules, some polyphenols, terpenes, and saponins have multiple and overlapped targets in other neurodegenerative diseases, including Alzheimer’s disease as well as PD [173,174,175]. Natural compounds may serve as preventive supplements for age-related neurodegenerative diseases, and can be applied in combinatorial treatments to improve the quality of life of patients. Natural compounds have been widely tested in α-synuclein- or neurotoxin-induced PD models. However, studies testing natural compounds for therapeutic purposes may have a limitation in terms of the differences of experimental design such as the quality of the extracts and the forms of dosage [176]. This could significantly affect the efficacy and toxicity of the natural compounds tested in each setting. Thus, it is necessary to organize the design of tests of natural compounds in PD models. The main limitation is the unclear therapeutic mechanism of natural compounds. These lead compounds can be adopted to design synthetic derivatives, but intensive study is required for further drug development. 

Although the bioavailability of the compounds from natural products is limited, they can be easily obtained from herbs, fruits, and marine organisms, and their intake is relatively safe, particularly via foods. Some extracts allow for the continuous absorption of multiple compounds at low doses over a lifetime, potentially evoking hormesis signaling, which may extend lifespans. Thus, further study is necessary.

## Figures and Tables

**Figure 1 ijms-20-05208-f001:**
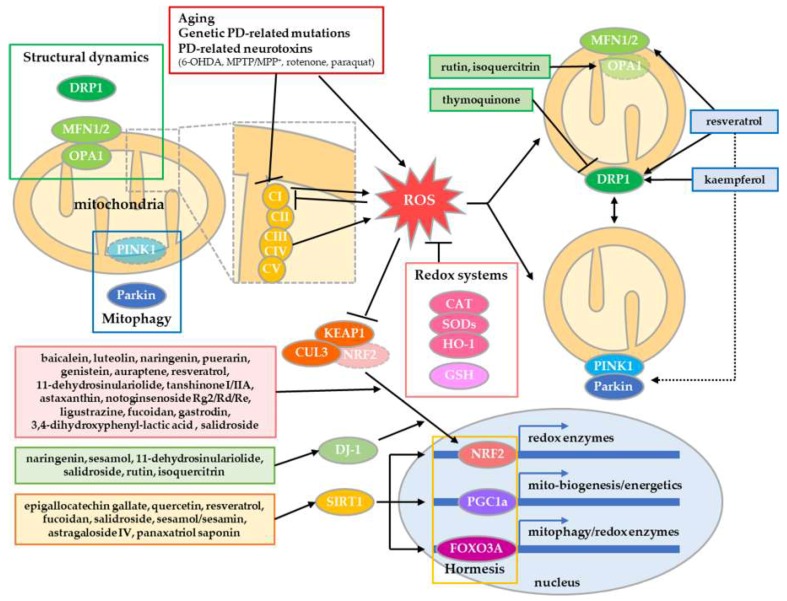
Neuroprotective compounds via mitochondrial quality control in PD. Mitochondrial quality is controlled by redox systems, structural dynamics, and mitophagy. In addition, it can be enhanced by hormetic adaptive stress responses. Some natural compounds revert and/or enhance redox system by NRF2 signaling, and improve mitochondrial quality by controlling structural dynamics and mitophagy. In addition, some compounds evoke adaptive stress responses mediated by SIRT1, which induce gene expression involved in redox enzymes, mitochondrial biogenesis/energetics and mitophagy. Therefore, these compounds protect DA neurons in PD.

**Figure 2 ijms-20-05208-f002:**
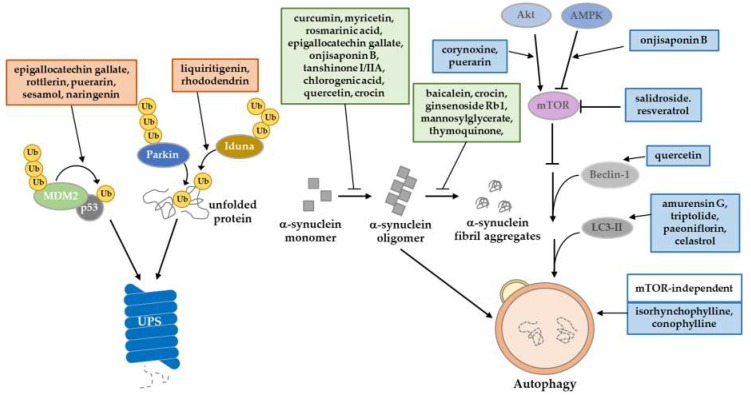
Summary of natural product regulation in proteostasis machinery. Natural products have a potential role to play in the amelioration of PD-induced proteostasis impairment. They regulate UPS through E3 ligase activity, increasing the autophagy‒lysosome pathway, and inhibiting the posttranslational modifications of α-synuclein.

**Table 1 ijms-20-05208-t001:** Lead compounds from natural products having neuroprotective effect in experimental PD model.

Class	Compounds	MitoQC	ProteoQC	Refs.	Class	Compounds	MitoQC	ProteoQC	Refs.
polyphenol/flavonoid	Epigallocatechin gallate	○	○	[61,62]	terpene/diterpene	11-Dehydrosinulariolide	△	△	[47]
	Apigenin	○	○	[63,64]		Tanshinone I	○	○	[48,65]
	Baicalein	○	○	[40,66]		Tanshinone IIA	○	○	[49,65]
	Luteolin	○	○	[41,67]		Triptolide		○	[68]
	Naringenin	○	○	[42,59]	terpene/triterpene	Celastrol	○	○	[69,70]
	Puerarin	○	○	[43,71]		Ursolic acid	○		[72]
	Quercetin	○	○	[73,74]		Asiaticoside A	○		[75]
	Rutin	○	△	[60,76]	terpene/sesquiterpene	Nerolidol	○		[77]
	Isoquercitrin	○	△	[60]	saponin	Astragaloside IV	○	○	[78,79]
	Kaempferol	○	△	[80,81]		Gypenosides	○		[82]
	Isoliquiritigenin	○		[83]		Notoginsenoside Rg1	○		[51]
	Genistein	○		[44]		Panaxatriol saponin	○		[84]
	Biochanin A	○		[85]		Onjisaponin B	△	○	[86]
	Hesperidin	○		[87]		Ginsenoside Rb1		○	[88]
	Morin	○		[89]		Ginsenoside Rd	○		[52]
	Myricetin	○		[90,91]		Ginsenoside Re	○		[52]
	Dihydromyricetin	○	○	[92,93]		Ginsenoside Rg1	○	○	[94,95]
	Troxerutin	○		[96]	alkaloid	Ligustrazine	○	○	[53]
	Liquiritigenin	○	○	[97]		Isorhynchophylline	○	○	[98,99]
polyphenol/coumarin	Auraptene	○		[45]		Conophylline	△	○	[100]
	Fraxetin	○		[101]		Amurensin G		○	[102]
	Esculin	○		[103]		6-Hydroxy-N-acetyl-β-oxotryptamine	○		[104]
	Esculetin	○		[105]	diketo-piperazine	Mactanamide	○		[104]
polyphenol/cinnamate	Chlorogenic acid	○	○	[106,107]	polyketide	8-Methoxy-3,5-dimethylisochroman-6-ol	○		[104]
	Curcumin	○	○	[90,108]		3-O-Methylorsellinic acid	○		[104]
	Rosmarinic acid	○	○	[109,110]	dibenzofuran	Candidusin A	○		[104]
polyphenol/stilbene	Resveratrol	○	○	[46,111,112,113]		4″-Dehydroxycandidusin A	○		[104]
	Piceatannol	○		[105]	mannose	Mannosylglycerate		○	[114]
	2,3,5,4′-tetrahydr-oxystilbene-2-O-β-D-glucoside	○	○	[115,116]	deoxy-adenosine	Cordycepin	○		[117]
	Salvianolic acid A	○	△	[118,119]	polysaccharide	Sulfated hetero-polysaccharides	○		[120]
	Salvianolic acid B	○	○	[93,121]		Sulfated galactofucan polysaccharides	○		[120]
	Polydatin	○		[122]		Fucoidan	○	△	[54,123,124]
polyphenol/xanthone	Mangiferin	○		[125]	quinone	Thymoquinone	○	○	[126,127]
polyphenol/lignan	Sesamol	○	○	[59,128]		2-methoxy-6-acetyl-7-methyljuglone	○		[129]
	Sesamin	○		[128]	anisole	β-asarone	○	○	[130]
	Magnolol	○		[131,132]	benzofurans	3-n-butylphthalide	○	○	[133]
terpene/carotenoid	Crocetin	○	○	[134,135]	glucoside	Gastrodin	○		[55]
	Crocin	○	○	[135,136]	bibenzyl	Chrysotoxine	○		[137]
	Astaxanthin	○	○	[50,138]	indolizine	Corynoxine B	△	○	[139]
terpene/monoterpene	Paeoniflorin	○	○	[140,141]	iridoid	Oleuropein	○		[76]
	Catalpol	○		[142]	lactate	3,4-dihydroxyphenyl-lactic acid	○		[56]
	Isoborneol	○		[143]	phenol-glycoside	Salidroside	△	○	[57,144,145]

We list the lead compounds in natural products having a neuroprotective effect in PD, and summarize their effects according to mitochondrial quality control (MitoQC) and protein quality control (ProteoQC), with references. Open circles or triangles indicate the existence of direct or indirect evidence in the literature, respectively.

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
