# Peer review of "Amelioration of Mitochondrial Quality Control and Proteostasis by Natural Compounds in Parkinson’s Disease Models"

_ijms, 2019, doi:10.3390/ijms20205208_

Round 1
Reviewer 1 Report
This review addresses the potential therapeutic benefits of natural compounds for Parkinson's disease. Several, mostly in vitro but also in vivo models are listed where these compounds could ameliorate mitochondrial function or proteostasis.
The review is concisely written, correctly describes PD pathogenesis and its molecular underpinnings. It lists a wealth of compounds where a specific molecular pathway can be linked to the mechanism of action. The structure is nicely organized into mitochondrial and proteostatic defects that can be ameliorated by naturally occuring molecules.
The Table and the Figures support the understanding of the diversity of compounds and the molecular pathways they are acting on.
Nonetheless, there are few minor issues that remain to be addressed.
Most importantly, Chapter 3.1 requires a thorough proofreading of English. Within that or in the Introduction, a better explanation of the function of p53 in PD pathomechanism is desirable - the description of the p53 degradation pathway would be meaningless without this. Hormesis is mentioned but not formally described - it would be good to read a few sentences about this dose-response phenomenon.
More minor points:
At line 81, mdivi-1 is not defined at all line 162 - mitophagy, not "mitophay " lines 259-261 - It is stated that " Some studies have reported that natural products can elevate autophagy by mediating a-synuclein degradation through the modulation of Akt/AMPK/mTOR signaling. " It is misleading to put it this way. Most likely, the natural products induce a-Syn degradation by elevating autophagic activity but this elevation is not mediated by a-Syn degradation, it is direct.Altogether, it is a nice manuscript that would have a wide readership once published.
Author Response
Comment 1>
This review addresses the potential therapeutic benefits of natural compounds for Parkinson's disease. Several, mostly in vitro but also in vivo models are listed where these compounds could ameliorate mitochondrial function or proteostasis.
The review is concisely written, correctly describes PD pathogenesis and its molecular underpinnings. It lists a wealth of compounds where a specific molecular pathway can be linked to the mechanism of action. The structure is nicely organized into mitochondrial and proteostatic defects that can be ameliorated by naturally occuring molecules.
The Table and the Figures support the understanding of the diversity of compounds and the molecular pathways they are acting on.
Response 1>
We thank the reviewer for interest in the manuscript and constructively addressed the concerns to improve in the revised manuscript additionally. We hope that the below given responses may satisfy the reviewer’s expectation.
Comment 2>
Nonetheless, there are few minor issues that remain to be addressed.
Most importantly, Chapter 3.1 requires a thorough proofreading of English. Within that or in the Introduction, a better explanation of the function of p53 in PD pathomechanism is desirable - the description of the p53 degradation pathway would be meaningless without this.
Response 2>
We are sorry for insufficient explanation, and included more detail function and relevance of p53 in PD experimental models in Chapter 3.1. (at line 214-220)
Comment 3>
Hormesis is mentioned but not formally described - it would be good to read a few sentences about this dose-response phenomenon.
Response 3>
In this revised manuscript, we added the description about hormesis and toxic dose-response of some compounds with a new references at line 176-178 and 189-191.
More minor points:
Comment 4>
At line 81, mdivi-1 is not defined at all line 162 - mitophagy, not "mitophay " lines 259-261 - It is stated that " Some studies have reported that natural products can elevate autophagy by mediating a-synuclein degradation through the modulation of Akt/AMPK/mTOR signaling. " It is misleading to put it this way. Most likely, the natural products induce a-Syn degradation by elevating autophagic activity but this elevation is not mediated by a-Syn degradation, it is direct.
Response 4>
In this revised manuscript, we defined and modified as per the reviewer’s suggestion. (at line 81, line 163, line 242-243)
Altogether, it is a nice manuscript that would have a wide readership once published.
We thank the reviewer’s helpful comments again.
Reviewer 2 Report
Dear Editor, I carefully read the manuscript ijms-621377, which regards an interesting topic. Some comments for the Authors: - English language needs to be carefully revised in order to correct the typos - References should be updated - Authors should cite the article doi: 10.1016/j.phrs.2017.12.029 - Authors should clarify which are the main limitations of the studies testing natural compounds in Parkinson's Disease models - Table A is extremely dispersive. Authors should group the compounds by class.Author Response
Point 1>
- English language needs to be carefully revised in order to correct the typos
Response 1>
We are sorry for inconvenient reading, and double-checked to correct the typo.
Point 2>
- References should be updated
Response 2>
In this revised manuscript, references style was updated as containing the article doi. ('References' section)
Point 3>
- Authors should cite the article doi: 10.1016/j.phrs.2017.12.029
Response 3>
We thank the reviewer for constructive suggestion, and the article was included in this revised manuscript. (Reference #119, at line 282 and 659-660)
Point 4>
- Authors should clarify which are the main limitations of the studies testing natural compounds in Parkinson's Disease models
Response 4>
In this revised manuscript, we addressed to clarify the limitation of natural compounds in ‘Conclusion and Future Prospects’ section. (at line 279-288)
Point 5>
- Table A is extremely dispersive. Authors should group the compounds by class.
Response 5>
We are sorry for inconvenience, table A was modified as grouping the compounds.
The manuscript was revised as per the reviewer’s suggestions. We hope that the revised manuscript will fulfill the reviewer’s comments